# Targeted and Untargeted Metabolic Profiling to Discover Bioactive Compounds in Seaweeds and Hemp Using Gas and Liquid Chromatography-Mass Spectrometry

**DOI:** 10.3390/metabo11050259

**Published:** 2021-04-22

**Authors:** Natalja P. Nørskov, Annette Bruhn, Andrew Cole, Mette Olaf Nielsen

**Affiliations:** 1Department of Animal Science, Aarhus University, Blichers Alle 20, 8830 Tjele, Denmark; mon@anis.au.dk; 2Center for Circular Bioeconomy, Aarhus University, 8830 Tjele, Denmark; 3Department of Bioscience, Aarhus University, Vejlsøvej 25, 8600 Silkeborg, Denmark; anbr@bios.au.dk; 4Center for Macroalgal Resources and Biotechnology, James Cook University, Douglas, QLD 4811, Australia; andrew.cole3@jcu.edu.au

**Keywords:** metabolomic profiling, seaweeds, hemp, phenolics, halomethanes, methane, Liquid Chromatography-Mass Spectrometry (LC-MS), Gas Chromatography-Mass Spectrometry (GC-MS), greenhouse gasses

## Abstract

Greenhouse gas emissions are a global problem facing the dairy/beef industry. Novel feed additives consisting of seaweeds and hemp containing bioactive compounds are theorized to reduce enteric methane emissions. In this study we aimed to investigate the metabolic profiles of brown, red and green seaweeds and hemp using gas chromatography and liquid chromatography mass spectrometry. We used targeted and untargeted approaches, quantifying known halomethanes and phenolics, as well as identifying potentially novel bioactive compounds with anti-methanogenic properties. The main findings were: (a) *Asparagopsis taxiformis* contained halomethanes, with high concentrations of bromoform (4200 µg/g DW), six volatile halocarbons were tentatively identified; (b) no halomethanes were detected in the other studied seaweeds nor in hemp; (c) high concentrations of lignans were measured in hemp; (d) a high numbers of sulfated phenolic acids and unidentified sulfuric acid-containing compounds were detected in all seaweeds; (e) flavonoid glucosides and glucuronides were mainly identified in hemp; and (f) the condensed tannin gallocatechin was tentatively identified in *Fucus* sp. Using the combined metabolomics approach, an overview and in-depth information on secondary metabolites were provided. Halomethanes of *Asparagopsis* sp. have already been shown to be anti-methanogenic; however, metabolic profiles of seaweeds such as *Dictyota* and *Sargassum* have also been shown to contain compounds that may have anti-methanogenic potential.

## 1. Introduction

Global warming and climate change due to the emission of greenhouse gasses (GHG) are important problems facing our world today, and emissions due to the enteric production of the GHG methane is attracting attention in this respect. Novel feeds and feed additives have been introduced in an attempt to markedly reduce enteric methane emissions in order to be able to comply with climate goals defined by the EU and the Danish government, as well as the dairy/beef industry worldwide [1]. A diverse range of anti-methanogenic feed additives are currently under investigation, including marine macroalgae and hemp, and new products are continuously being proposed [1].

Many chemical feed additives have been investigated and are still under investigation. However, during the recent decade a paradigm shift has been initiated in order to find natural sources of safe feed additives as alternatives to chemical additives to inhibit enteric methanogenesis [2,3]. A variety of natural sources, mainly extracts from different plant species, have been screened for anti-methanogenic activity. For example, Bodas et al. [4] screened more than 450 plant species and reported a 15% methane reduction caused by six different plants. Recently, a new source of anti-methanogenic activity was discovered—seaweeds [5,6,7,8]. The anti-methanogenic activity of terrestrial plants is attributed to plant secondary metabolites such as tannins, saponins and flavonoids [3,9]. In seaweeds the anti-methanogenic activity is mainly associated with halomethanes [1,10]. In vivo experiments have shown that the red seaweed species *Asparagopsis taxiformis* and *Asparagopsis armata* efficiently inhibit ruminant microbial methanogenesis [11,12]. The inclusion of 0.1% and 0.2% *Asparagopsis taxiformis* in the feed of Brahman–Angus cross steers reduced methane emissions by 40% and 98% and demonstrated weight gain improvements of 53% and 42%, respectively [11]. In that study, no negative effects on feed intake, feed conversion efficiencies nor on rumen function or meat eating quality were detected. The anti-methanogenic effect was attributed to halogenated compounds such as bromoform, dibromochloromethane and dibromethane [10,13]. The toxicity and ozone degrading properties of halocarbons are well known, and there are both health and environmental concerns related to the use of *Asparagopsis taxiformis* as a feed additive [14]. Brown seaweeds have also shown anti-methanogenic potential [5,6]. Previous in vitro studies have indicated that brown seaweeds such as *Dictyota bartayresii* and *Zonaria farlowii* have methane-reducing potential [5,6]. *Dictyota bartayresii* has been shown to inhibit the production of methane by over 92% [5]. The methane-reducing effects of seaweeds other than *Asparagopsis* are attributed to secondary metabolites produced by and presumably specific to the species; however, the exact nature of these compounds has not yet been identified. Seaweeds produce and contain a range of other bioactive compounds, including phenolic compounds such as flavonoids, phlorotannins, phenolic acids and some derivatives of lignans [15,16].

*Cannabis sativa* L., known as hemp, is a rich source of secondary metabolites and contains phenolics such as flavonoids, stilbenes and phenolic acids, and recently it has been discovered that seeds of hemp also contain low concentrations of lignans [17]. Organic hemp cultivation is increasing in Denmark and in the whole of Europe, and hemp-derived products have found applications in textiles, construction materials, cosmetics and the medical industry, as well as in food and feed supplementation [18]. With the growing acceptance of hemp cultivation, there has been speculation as to whether hemp may be an acceptable source of feed supplements for ruminants. Therefore, seaweeds and terrestrial plants with profiles high in secondary metabolites are presently subject to considerable attention as potential new feed supplements in ruminant production in order to achieve the goal of methane reduction; however, more knowledge is needed on the concentration and presence of known and potentially novel metabolites that will enter the food chain.

Flavonoids and tannins are diverse classes of phenolic compounds with anti-microbial activity. Anti-methanogenic effects of tannins have been extensively studied in several in vitro and in vivo studies [3]. Methane production from ruminal fermentation has been decreased by up to 50% in response to tannin-rich forage or plant extracts containing these phenolic compounds in vitro [3]. Although tannins are recognized as compounds with high methane-reduction potential, not all studies support this fact [2,3]. There is a general agreement in the literature that the molecular weight of tannins is of importance in this respect, and that the anti-methanogenic effect is more pronounced for condensed tannins with high molecular weights [2,3]. The mechanisms of their action are less understood, however. It appears that condensed tannins are capable of reducing methane production through reduced fiber digestion, whereas low-molecular-weight hydrolysable tannins seem to act through inhibition of growth or the activity of rumen methanogens [1,2,3]. Flavonoids have not been as well studied as tannins; however, in vitro screening studies with different plant species rich in flavonoids have reported up to 15% methane reductions, with no adverse effects on feed digestibility [3]. Their proposed mechanisms are similar to those of tannins, and flavonoids can inhibit the growth of methanogens and also act as H_2_ sinks via the cleavage of ring structures and reductive dehydroxylation [1].

The aim of the present study was to: (a) identify and quantify the profile of halomethanes in seaweeds and hemp, (b) identify the metabolic profile of seaweeds and hemp and identify potentially novel anti-methanogenic volatile and non-volatile metabolites/phenolics, and (c) apply a targeted approach to quantify 15 phenolic compounds in seaweeds and hemp. Moreover, we hypothesized that the combination of targeted and untargeted approaches would provide the best overview and in-depth information on secondary metabolites present in seaweeds and hemp from the perspective of methane mitigation.

## 2. Results

### 2.1. Targeted and Untargeted GC-MS Analyses

Extracts designated as “Extract 1” from seaweeds and hemp were subjected to GC-MS analyses for the quantification of bromoform, dibromomethane and dibromochloromethane using SIM mode and for the metabolic profiling of volatile halocarbons using full scan mode. Halomethanes and other volatile halocarbons were only detected in red seaweed *Asparagopsis taxiformis* (Figure 1a) and not in any of the other studied seaweed species nor in the hemp samples analyzed (Figure 1b). *Asparagopsis taxiformis* contained three halomethanes—bromoform (4200 µg/gdry weight), dibromomethane (1.0 µg/g dry weight) and dibromochloromethane (140 µg/g dry weight). Other halomethanes and volatile halocarbons that were tentatively identified in the full scan of Extract 1 of *Asparagopsis taxiformis* were acetic acid dichloromethyl ester, dibromonitromethane, tetrachloroacetone, tetrabromomethane, 4-(4-bromophenyl) pyridine and bromodiiodomethane (Figure 1a).

### 2.2. Untargeted LC-MS Analyses

Extracts designated as “Extract 2” from seaweeds and hemp were subjected to untargeted LC-MS analyses for the metabolic profiling of red, green and brown seaweeds and hemp, as well as for identification of potentially novel compounds/phenolics (Figure 2). Using MS/MS fragmentation spectra and neutral loss patterns, combined with monoisotopic masses of high resolution mass spectrometry and online databases, untargeted LC-MS analysis was able to provide an overview of potential compounds present in the samples. Because phenolic compounds have hydrophobic properties, reverse-phase chromatography with a C_18_ column was the method of choice. The gradient of chromatographic separation started with a low concentration of organic solvent, which promoted elution of less hydrophobic molecules with low molecular weights. As the gradient increased with the concentration of organic solvent, more hydrophobic molecules were eluted. In general, the metabolic profiles of seaweeds, as shown in Figure 2, were dominated by the elution of metabolites in the beginning of the chromatogram at 0.5 to 5 min (5% to 25% acetonitrile) and then at 9.5 to 20 min (50% to 100% acetonitrile), whereas the metabolic profiles of hemp were markedly different, with metabolites eluting throughout the gradient.

The metabolic profiles of seaweeds, except for *Asparagopsis taxiformis*, were dominated by spectra/compounds containing sulfuric acid H_2_SO_4_ (96.9601 *m*/*z*), hydrogen sulfite HSO_3_^−^ (79.9574 *m*/*z*), sulfite H_2_SO_3_ (80.9652 *m*/*z*) and mesylic acid CH_4_SO_3_ (94.0908 *m*/*z*). A typical neutral loss of 80 Da, corresponding to the loss of sulfonic acid SO_3_, was highly abundant, along with neutral losses of 44 and 46 Da for carboxylic acid CO_2_ and CH_2_O_2_, respectively. Phenolic compounds detected and identified in the chromatograms could mainly be assigned to phenolic acids and their derivatives. The compounds identified were tyrosol 4-sulfate, dihydrocaffeic acid 3-sulfate, phenylpropanoic acid sulfate, tyrosin sulfate, ethyl gallic acid 3-sulfuric acid, vanillic acid 4-sulfate, vanillin sulfate, ethyl hydrogen disulfate and monosulfate, *p*-hydroxybenzoic acid, *p*-hydroxybenzaldehyde, benzoic acid, benzenesulfonic acid and phenyl sulfates in general. They were tentatively identified using autoMS/MS fragmentation and HMDB. Phenolic acids such as *p*-hydroxybenzoic acid with precursor ion *m*/*z* 137.0295 and fragment *m*/*z* 93.0345 and *p*-hydroxybenzaldehyde with precursor ion *m*/*z* 121.0295 and fragment *m*/*z* 92.0269 were identified in the red seaweed *Delesseria sanguinea*. Vanillic acid 4-sulfate or 4-methoxy-3-(sulfooxy)benzoic acid with *m*/*z* 246.9922 and the loss of carboxylic acid-generating fragment *m*/*z* 203.0013 was tentatively identified in *Ulva intestinalis* (Figure 3). High intensity ions *m*/*z* 297.1375 and *m*/*z* 251.1318 containing sulfuric acid were detected but not identified in brown seaweed species such as *Sargassum muticum* and *Dictyota dichotoma* (Figure 3). The metabolic profile of *Asparagopsis taxiformis* was dominated by brominated, chlorinated and iodine-containing compounds. Bromine occurs naturally as two isotopes, ^79^Br and ^81^Br, with relative abundances of 1:1 (50.69%:49.31%) respectively, whereas chlorine occurs naturally as ^35^Cl and ^37^Cl with relative abundances of 3:1 (75.77%:24.23%) [19]. Both isotopes of Br were detected in MS and MS/MS spectra. The characteristic clusters of M+2, M+4, M+6, etc. [19] with varying isotopic patterns were detected throughout the chromatogram, indicating that up to four halogen atoms were present in the molecules (Figure 3). In addition to *Asparagopsis taxiformis*, the brown seaweed *Dictyota dichotoma* contained brominated and chlorinated compounds, whereas only a few chlorinated compounds were detected in the other red, brown and green seaweeds, *Sargassum muticum*, *Gracilaria vermiculophylla*, *Laminaria digitata*, *Saccharina latissima*, *Ulva* sp. and *Ulva intestinalis*. Compounds containing iodine *m*/*z* 126.9045 were highly present in *Asparagopsis taxiformis* and were also detected in two brown seaweed species, *Saccharina latissima* and *Laminaria digitata*. A high number of glycolipids, polyunsaturated fatty acids (PUFAs) and derivatives thereof with the neutral losses of 197 Da and 282 Da (oleic acid), eluting after approximately 13.5 min, were observed in the red seaweed *Delesseria sanguinea*, the brown seaweeds *Sargassum muticum* and *Dictyota dichotoma* and the green seaweeds *Ulva* sp. and *Ulva intestinalis*. Several mono- and diterpenoid glucosides were detected in *Delesseria sanguinea*, *Sargassum muticum*, *Gracilaria vermiculophylla* and *Fucus* sp. Ascorbic acid was highly abundant in *Fucus* sp., and the disaccharide melibitol and the bioflavonoid gallocatechin were also present in these species. The metabolic profile of brown seaweeds contained mannitol, a characteristic sugar alcohol, which was eluted at the beginning of the chromatograms. The metabolic profile of hemp contained a high number of characteristic conjugation patterns for flavonoid glucuronides and glucosides, with neutral losses of 162 Da for hexoside, 146 Da for deoxyhexoside, 120 Da for glucoside and 176 Da for glucuronide [20]. Among the identified metabolites were kaempferol-3-O-glucoside, apigenin-7-O-glucoside, luteolin-7-O-glucoside, rutin, chlorogenic and p-coumaric acids and azelaic acid. Cannabinoids such as cannabisin A and tetrahydrocannabinol (THC) were also tentatively identified (Figure 4).

In general, the metabolic profile of *Delesseria sanguinea* contained the highest number of detected metabolites among the red seaweeds analyzed, whereas *Sargassum muticum* and *Dictyota dichotoma* contained the highest number of detected metabolites among the brown seaweeds. By far the highest number of detected metabolites and phenolics were detected in *Cannabis sativa* L., which was eluted between 2.5 and 12.5 min (Figure 2).

### 2.3. Targeted LC-MS/MS Analyses

Extracts 2 and 3 of seaweeds and hemp were subjected to targeted LC-MS/MS analyses for the quantification of lignans (matairesinol, hydroxymatairesinol, secoisolariciresinol, lariciresinol, isolariciresinol, syringaresinol, medioresinol, pinoresinol) and isoflavones (naringenin, formononetin, chrysin, genistein, daidzein, glycitein and prunetin). No lignans were detected through targeted LC-MS/MS analysis in any of the seaweeds analyzed. Only low concentrations (LLOQ) of isoflavones daidzein and genistein were detected in two seaweed species, *Saccharina latissima* and *Ulva intestinalis*, and formononetin was detected in *Saccharina latissimi* and *Fucus vesiculosus*. In comparison, hemp contained high concentrations of lignans (Table 1) and two isoflavones, naringenin and glycitein, were also detected.

When analyzing Extract 2, free forms of lignans and isoflavones were measured in the methanolic extraction of freeze-dried material. When analyzing Extract 3, the lignan and isoflavone glycosides, glucuronides and sulfates were released by β-glucuronidase/sulfatase and total concentrations were subsequently able to be measured. Lariciresinol, syringaresinol, isolariciresinol and naringenin contents were highest in *Cannabis sativa* L. variety Futura 75, whereas secoisolariciresinol was highest in *Cannabis sativa* L. variety Finola. Matairesinol showed a low concentration, as did glycitein, in both varieties. A representative chromatogram of *Cannabis sativa* L. Extract 3 is presented in Figure 5.

## 3. Discussion

Due to the increasing interest in feed-based methane mitigation strategies for ruminant livestock, seaweeds and hemp are attracting more and more attention as they can fulfil several sustainability goals, being fast-growing, high-biomass-yielding crops, which are rich in dietary fibers and secondary metabolites, with some of them also being rich in proteins [7,21]. However, the profiles of secondary metabolites vary greatly between terrestrial plants, such as hemp, and marine organisms, such as seaweeds, and these also vary within seaweeds, depending on whether they are red, brown or green seaweeds.

Our targeted GC-MS analyses showed that *Asparagopsis taxiformis* contained high concentrations of the halomethane bromoform, and dibromomethane and dibromochloromethane were also quantified. In contrast, no halomethanes were detected in any of the other studied red, brown or green seaweeds species, nor in hemp stem or leaves. Previous studies on the release of volatile halocarbons from seaweeds have shown the release of bromoform and other halomethanes from some spices of brown, red and green seaweeds [13]. Therefore, it cannot be excluded that some of the seaweeds in this study may have contained low concentrations of halomethanes, which were below the detection limits of our GC-MS system. High concentrations of bromoform have been quantified in *Asparagopsis* sp. in previous studies. In a study by Machado et al. [10], the bromoform content of *Asparagopsis taxiformis* was 1723 µg/g dry matter (DM), whereas Alisa Mihaila [22] reported contents of bromoform of *Asparagopsis armata* varying from 1.0 to 10.4 mg/g DM due to locational differences. This indicates that our result of 4200 µg/g DM is in accordance with the published literature. The untargeted GC-MS analysis identified a high signal from a compound tentatively identified as dibromonitromethane. To our knowledge, there are no previous records of this compound in *Asparagopsis* sp. Furthermore, our untargeted LC-MS analyses provided support to the results of the targeted and untargeted GC-MS analyses, showing that the metabolic profiles of *Asparagopsis taxiformis* contained mainly halogenated compounds with varying numbers of Cl and Br atoms. It has been previously reported that *Asparagopsis* sp. contained more than 100 halogenated compounds—haloforms, haloacids and haloketons [23,24]. Halogenated compounds, and bromoform in particular, have been shown to be extremely effective anti-methanogens and different methanogenesis-inhibiting mechanisms have been proposed [1]. Although in this study detectable contents of halomethanes were only found in *Asparagopsis taxiformis*, we did find brominated and iodine-containing compounds in several other studied seaweeds, *Dictyota dichotoma, Saccharina latissima* and *Laminaria digitate*. These compounds were detected by untargeted LC-MS with characteristic spectra of M+2 etc. ions eluting in the beginning of the chromatograms. However, the search for their identification in HMDB was not successful. Previous studies have identified bromophenols in brown seaweeds and halogenated monoterpenes in red seaweeds [7,25]. It has been reported that the contents of Br and I were the highest in *Asparagopsis armata* as compared to other red, brown and green seaweeds [22]. However, compared to Cl, the contents of Br and I were low [22]. It is reasonable to assume, however, that the high content of Cl in seaweeds was associated with salt-water content.

Organic sulfur compounds are known to be essential for organismal survival, and the sulfur cycle in the ocean has now been well described [26]. It is also known that high concentrations of organosulfur compounds are produced and stored by micro- and macroalgae [26]. Sulfation and desulfation reactions, known as the sulfation pathway, modify the variety of secondary metabolites in plants and animals [27]. Sulfotransferases are a large family of enzymes with different substrate specificities that conjugate a plethora of metabolites; however, the function of this conjugation is not well understood [27]. The metabolic profiles of seaweeds in this study revealed a high number of sulfated metabolites belonging to phenolic acids and derivatives. Similarly to previous studies using high resolution mass spectrometry [15], both hydroxybenzoic and hydroxycinnamic acids were identified. In a study by Zhong et al. [15] vanillic acid 4-sulfate was identified in the green seaweed *Ulva* sp. and brown *Sargassum* sp. In our study, vanillic acid 4-sulfate was also detected in *Ulva intestinalis*, but not in *Sargassum muticum*. Moreover, in a study by Zhong et al. [15], hydroxytyrosol 4-O-glucoside was identified in *Sargassum muticum*, whereas in our study tyrosol sulfate was only identified in the red seaweed *Delesseria sanguinea*. Tyrosol conjugates have been previously reported in red seaweeds [15], but not in *Delesseria sanguinea*. Zhong et al. [15] also identified *p*-hydroxybenzoic acid and *p*-hydroxybenzaldehyde in red, brown and green seaweeds. We identified these compounds in *Delesseria sanguinea*. Phenolic acids are low-molecular-weight compounds that are regularly bound to complex carbohydrates, which then attain a complex structure, and cell wall rigidity is thereby controlled by carbohydrate structure and phenol cross-linking [8]. Phenolic acids are known for their antioxidative and antimicrobial properties and have also been shown to reduce methane production [8]. These compounds are also found in high concentrations in cereals such as wheat and maize, with ferulic acid being the main phenolic acid component in the cross-linking of arabinoxylan, the main cereal cell wall polysaccharide [28]. However, phenolic acids in cereals are known to be present mainly as glycoside or glucuronide conjugates [28,29], and our results indicate that the opposite is observed in seaweeds. Furthermore, sulfate itself and sulfur-containing compounds, known as organosulfur compounds, have also been shown to reduce methane production [1]. Unidentified sulfuric acid-containing compounds in metabolic profiles of *Sargassum muticum* and *Dictyota dichotoma* may therefore be of interest with respect to anti-methanogenic properties. In addition to sulfated phenolic acids, seaweeds contain sulfated cell wall polysaccharides, sulfated galactanes and sulfated glycolipids [7,8,30]. Adversely, high concentrations of sulfate may have negative effects on feed intake and animal performance and may increase the risk of sulfur-associated polioencephalomalacia [1,2]. Therefore, further attention should be given to the contents of sulfate and sulfur in seaweeds in relation to animal feeds.

Flavonoids and lignans were other groups of phenolics assessed in this study, and were mainly identified and subsequently quantified in the *Cannabis sativa* L. variety Futura 75 and Finola. Previous studies of *Cannabis sativa* L. leaves, stem and inflorescence have shown similar metabolic profiles, with high contents of flavonoids and cannabinoids [31,32]. To our knowledge, this is the first study to identify and quantify the concentration of lignans in the top part (the mixture of stem and leaves) of the hemp plant. Previously, Smeds et al. [17] quantified lignans in the seeds, and found that they were dominated by syringaresinol, medioresinol, secoisolariciresinol, lariciresinol and pinoresinol. Our targeted LC-MS analyses also showed secoisolariciresinol, lariciresinol, syringaresinol and isolariciresinol to be the predominant lignans, although we did not detect any medioresinol and pinoresinol. Higher concentrations of lignans were quantified in the study of Smeds et al. [17] compared to those of our study, indicating that the concentration of lignans is higher in seeds compared to the stalk and leaves; however, the concentrations of lignans have high seasonal variations. In the current study, no lignans were detected in any of the seaweeds analyzed. This conforms with the literature, in which lignans have not previously been reported in seaweeds; however, derivatives of lignans were identified in seaweeds by Zhong et al. [15]. Our data support the view that lignans are much more abundant in terrestrial plants and play a limited role in seaweeds, that are phylogenetically distant from plants and do not have structural tissue. In the diet of ruminants, lignans are present in high concentrations in rye and wheat [17]. Isoflavones such as daidzein, genistein and formononetin were detected in a few seaweeds in low concentrations when performing targeted LC-MS analyses. The presence of daidzein and genistein has previously been found in red and brown seaweeds; however, formononetin has not been previously detected [25]. Daidzein and genistein are known as soy isoflavones, whereas formononetin, naringenin and glycitein are found in clover grasses [33,34]. These isoflavones are thereby regular constituents of the ruminant diet. Polyphenols represent an extremely diverse class of phenolic compounds with more than 10,000 representatives [35]. There are studies that support the presence of flavonoids, mainly as glucosides, in seaweed species [25]. In a study by Zhong et al. [15], 17 flavonoid glucosides, glucuronides and aglucones were tentatively identified in red, brown and green seaweeds. Brown seaweeds are known to contain the highest concentration of phenolics by DW (14%) compared to red and green seaweeds [36]. In this study, the metabolic profile of brown seaweed *Fucus* sp. revealed the presence of flavonoid glucosides, with one tentatively identified as gallocatechin (a condensed tannin). Zhong et al. [15] also identified gallocatechin in brown and green seaweeds. The effect of flavonoids on methanogenesis has not been well studied; however, one class of flavonoids, proanthocyanidins, known also as condensed tannins, has shown promising results in vitro and in vivo [2,3]. In a study by Milledge et al. [8], epicatechin (a condensed tannin), phloroglucinol (phlorotannin) and gallic acid (phenolic acid) were model phenolic compounds used in in vitro experiments simulating rumen fermentation. All three compounds showed the potential to reduce anaerobic digestion and the associated methane formation. In another in vitro study by Tan et al. [37], extracts of condensed tannins reduced methane production by 47%, with only a 7% reduction in feed DM [37]. However, high inclusion doses of condensed tannins have substantive negative effects on DM digestibility. Thus, the consideration of adequate inclusion doses is important in order to decrease enteric methane formation without adversely affecting rumen metabolism [2,36].

Aside from phenolics and halomethanes, high numbers of glycolipids and polyunsaturated fatty acids (PUFAs) were detected in the metabolic profiles generated through untargeted LC-MS of the red, brown and green seaweeds *Delesseria sanguinea, Sargassum muticum, Dictyota dichotoma* and *Ulva* sp., primarily eluting at the end of the chromatograms. Fatty acids and derivatives such as azelaic acid and oxylipins were identified in the metabolic profiles of hemp. Azelaic acid, which was identified in hemp, was also previously identified in wheat and rye [38]. In general, the lipid content of seaweeds is low, varying between 1% and 5% DM [7], but there are substantial differences between the species when studying the elution profiles using untargeted LC-MS. Hemp leaves and stems, on the other hand, have higher lipid contents, of 19.97% and 8%, respectively [39]. Studies with ruminants have shown that PUFAs in the diet can reduce methane emissions by acting as hydrogen sinks because biohydrogenation of PUFAs can compete with methanogenesis for metabolic H_2_ [40]. To our knowledge, no studies investigating glycolipids and methane reduction have been undertaken. Glycolipids are carbohydrate-attached lipids, with three major types biosynthesized in seaweeds: monogalactosyldiacylglycerides (MGDGs), digalactosyldiacylglycerides (DGDGs) and sulfoquinovosyldiacylglycerides (SQDGs) [30]. Their role is to provide energy and serve as markers for cellular recognition [36], whereas any possible impact on methanogenesis is still unexplored. In addition to lipids, we detected several tentatively identified terpenoids, which might also have the potential to inhibit methanogenesis [1,2].

Combining targeted and untargeted metabolic profiling evidently provided an extensive overview and in-depth knowledge about the existence and chemical nature of known and unknown metabolites in seaweeds and hemp. Clear metabolic differences were observed between seaweeds and hemp, as well as among red, brown and green seaweeds. Targeted metabolic GC-MS profiling provided evidence and confirmed the presence of high concentrations of halomethanes in *Asparagopsis taxiformis,* whereas halomethanes in other red, brown and green seaweeds were below the detection limit. Untargeted metabolic GC- and LC-MS profiles supported the targeted analyses and vice versa. No volatile halocarbons were detected in red, brown and green seaweeds except for *Asparagopsis taxiformis* in untargeted GC-MS analyses. New metabolites were identified in the metabolic profile of *Asparagopsis taxiformis* and spectrometric analyses of both GC- and LC-MS revealed high numbers of Br, Cl and I-containing compounds. Targeted LC-MS analyses demonstrated the presence of lignans in hemp plants, although lignans were absent from seaweeds. Untargeted metabolic LC-MS profiles revealed high numbers of sulfated phenolic acids. Targeted LC-MS detected low concentrations of isoflavones genistein and daidzein, and few flavonoids were also identified in the untargeted metabolic LC-MS profiles of seaweeds. However, flavonoids are an extremely heterogeneous group of compounds, with a high number of representatives that vary from species to species and, and of which the concentrations are influenced by seasonal and environmental changes, which were not taken into account in this study. The untargeted metabolic LC-MS profiles of hemp, on the other hand, showed high numbers of flavonoid glucosides, glucuronides and aglycones, which was also supported by the literature. In conclusion, our targeted and untargeted approaches confirmed that both seaweeds and hemp contain already known and well-examined compounds, such as phenolic acids and lignans, which are already present in the diets of ruminants. Conversely, novel metabolites such as halocarbons, condensed tannins, sulfated and unidentified sulfuric acid-containing compounds may enter the food chain, with unknown biological effects, some of which may account for anti-methanogenic properties. For example, the sulfated and unidentified sulfuric acid-containing compounds may be responsible for the anti-methanogenic properties of *Dictyota*. The flavonoids of *Fucus* sp. might also be responsible for the interest in these seaweeds as feed additives. Furthermore, metabolic profiling revealed the unique chemistry of *Asparagopsis taxiformis*, containing volatile halocarbons, which are known to have high methane-reductive potential.

## 4. Materials and Methods

### 4.1. Seaweed and Hemp Plant Samples

Eighteen species of seaweeds—seven red algae (Rhodophyceae), two green algae (Chlorophyceae) and nine brown algae (Phaeophyceae)—were procured from either wild harvest or cultivation (Table 2). Seventeen species were harvested in Nordic waters, whereas *Asparagopsis taxiformis*, a tropical red algae, was sampled in Australia to serve as a benchmark for the content of bromoforms. *Asparagopsis taxiformis* was collected by SCUBA divers from a depth of six meters at Nelly Bay (19.2° S, 146.8° E), Magnetic Island, Australia, on 8 September 2019. The harvested biomass was cleaned to remove contaminants and placed inside small ziplock bags. These bags were then covered with dry ice and transported to James Cook University and stored at −80 °C for 48 h. This biomass was then transferred to a freeze dryer until dry (0.012 Pa > 72 h). The lyophilized biomass was vacuum-sealed for final storage. The remaining species were harvested by snorkelling and SCUBA diving at depths down to three meters (*D. sanguinea* at 15 m) and either oven dried at 40 °C, then vacuum sealed for transportation (for samples cultivated and harvested at the Faroe Islands), or stored at −20 °C prior to freeze-drying at European Freeze Dry ApS, (Kirke Hyllinge, DK) (Table 2). All freeze-dried seaweed samples were milled to pass through a 1-mm screen and homogenized on an Ultra Centrifugal Mill ZM 200 (Retsch, Haan, Germany) (Figure 6).

Two varieties of hemp, *Cannabis sativa* L. variety Futura 75 and Finola, were harvested in Denmark on 24 June 2018 and 31 August 2019, respectively. The top part of the hemp plant (leaves and stalks) was harvested and dried for 5 days and seeds were threshed. After that, the dried hemp biomass was milled into smaller pieces. At Aarhus University, the hemp biomass was further milled to pass through a 1-mm screen and homogenized on an Ultra Centrifugal Mill ZM 200 (Retsch, Haan, Germany) (Figure 7).

### 4.2. Extraction of Seaweed and Hemp Samples for Gas Chromatography-Mass Spectrometry (GC-MS) and Liquid Chromatography-Mass Spectrometry (LC-MS) Analyses

Extract 1: Seaweed and hemp samples of 50 mg were weighted and added to 1.25 mL of methanol containing internal standards (IS) for GC-MS and LC-MS analyses, then mixed and sonicated for 15 min and vortexed for 1 h at room temperature. The samples were then centrifuged for 15 min at 20 °C at 14,000× *g*, and 200 µL of supernatant was transferred to a GC vial for GC-MS.

Extract 2: Supernatant (200 µL) from Extract 1 was diluted with 0.1% formic acid in water 1:3 supernatant:water and centrifuged for 10 min at 4 °C at 29,700× *g* and transferred to an HPLC vial for targeted and untargeted LC-MS.

Extract 3: Supernatant (800 µL) from Extract 1 was transferred to a new tube and evaporated to dryness under an N_2_ stream with a temperature of 60 °C. Hydrolysis of lignan and isoflavone glycosides was carried out by adding 0.8 mL of 0.05 NaOAc pH 5 containing freshly dissolved β-glucuronidase (≥300,000 units/g solid)/sulphatase (≥10,000 units/g solid) enzyme (2 mg/mL), following incubation in a shaker overnight at 37 °C according to the previously developed methods of Nørskov et al. [34,41]. After overnight incubation, samples were added to 200 µL acetonitrile containing 0.4% of formic acid and centrifuged for 15 min at 4 °C at 29,700× *g* and the supernatant was transferred to an HPLC vial for targeted LC-MS. Targeted analyses of samples were performed in duplicate.

### 4.3. Targeted and Untargeted GC-MS

#### 4.3.1. Materials

The following standards were used for quantitative GC-MS measurements: bromoform (certified reference material 5000 µg/mL in methanol), dibromomethane (analytical standard 1 g), bromochloromethane (certified reference material 200 µg/mL in methanol) and internal standard (IS) diiodomethane d2 (1 g) (Sigma, Merck KGaA, Darmstadt, Germany). Methanol was GC-MS grade, obtained from Sigma (Merck KGaA, Darmstadt, Germany). The stock standard solution was prepared through the dilution of purchased standards in methanol and stored at −20 °C when not in use.

#### 4.3.2. Standard Curve

The standard curve was prepared from the stock solution in methanol in the concentration range of 1–50 µg/mL with five points per compound quantified, and contained IS diiodomethane d2 in a concentration of 50 µg/mL. The concentration of each compound was calculated based on the peak area ratio of the targeted compound over the IS using the standard curve in Xcalibur 2.0.7 (Thermo Scientific, Waltham, MA, USA).

#### 4.3.3. GC-MS

GC-MS analyses were performed on a Trace GC ultra-coupled to a single quadrupole DSQ II mass spectrometer (Thermo Scientific, Waltham. MA, USA) in full scan (50–460 *m*/*z*) and selected ion monitoring (SIM) modes (Table 3). Two microliters of the sample (Extract 1) was injected in the split mode and separated on the Rxi-5ms capillary column (cross-linked 5% diphenyl/95% dimethylpolysiloxane, 30 m length, 0.25 mm I.D, 0.25 µm film thickness) (Restek, Bellefonte, PA, USA). The flow rate of the carrier gas (He) was 1.2 mL/min. The inlet and transfer line temperatures were 250 °C and 280 °C, respectively. The temperature program for the column was as follows: initial temperature, 40 °C (5 min hold), followed by a first ramp to 160 °C at 20 °C/min, and then a second ramp to 280 °C (4 min hold) at 60 °C/min. The data from the full scan were subjected to compound identification in Xcalibur 2.0.7 and PubChem.

#### 4.3.4. Method Validation

According to EU guidelines, trueness in terms of average recovery should be within the range of 70–120%, with the associated precision calculated as RSD ≤ 20%. The low limit of quantification (LLOQ) is defined as the lowest spiked concentration which meets the recovery and precision criteria [42]. Recovery experiments were performed in a representative matrix of seaweeds, which did not contain halomethanes. Recovery and precision were tested by spiking at three concentrations: low (1 µg/mL), medium (5 µg/mL) and high (50 µg/mL), with five replicates per concentration (Table 4). Recovery % was calculated as (measured value/spiked value)*100 and relative standard deviation (RSD) was calculated as standard deviation (SD)/average recovery*100. Reproducibility was also monitored by means of the peak shape, intensities and retention times of IS. Matrix effects were validated using the area under the curve of IS in the sample matrix divided by the area under the curve of IS.

### 4.4. Untargeted LC-MS

#### 4.4.1. Materials

Acetonitrile, isopropanol (VWR, West Chester, PA, USA) and formic acid (Fluka, Merck KGaA, Darmstadt, Germany) were LC-MS grade. The following IS was used: glycocholic acid (Glycin-1 ^13^C) from Sigma (Merck KGaA, Darmstadt, Germany). Lithium formate monohydrate 98% was purchased from Sigma (Merck KGaA, Darmstadt, Germany).

#### 4.4.2. LC-MS

Chromatographic separation was performed on an ultra-performance liquid chromatography (UPLC) Ultimate 3000 (Dionex, Sunnyvale, CA, USA) system, equipped with a Fortis C18 column, 100 × 2.1 mm, 1.7 µm (Fortis Technologies, Neston, UK). The column oven was set to 30 °C, the temperature of the autosampler was 5 °C and the flow was 400 µL/min. The solvent system consisted of solvent A (0.1% formic acid in water) and solvent B (0.1% of formic acid in acetonitrile). The gradient started at 5% acetonitrile and continued to 100% for 23 min with a post and pre-equilibration of 2 min, and 5 µL of sample was injected on the column. The UPLC system was connected to an impact HD quadrupole Time-of-flight (QTOF) mass spectrometer from Bruker Daltonics (Bremen, Germany), operated in full scan mode from 50 to 1000 *m*/*z* at a sampling rate of 1 Hz. The measurements were performed in negative ionization mode using electro spray ionization (ESI), which is known to promote the ionization of phenolic compounds. Capillary and end plate offset were set to −4500 and −500 V, respectively. The dry gas flow was set to 8 L/min, ion source temperature to 200 °C and nebulizer pressure to 1.8 bar. The collision energy during MS scanning was set to 6 eV. For MS/MS analyses, Ar was used as a collision gas and auto-MS/MS was performed with a collision energy between 10 and 40 eV. Samples were subjected to MS scanning for compound detection and auto-MS/MS scanning for compound identification. The identification of compounds was performed by comparing accurate mass and MS/MS spectra with the published literature and the online Human Metabolome Database (HMDB).

#### 4.4.3. Quality Assurance of the Method

The quality of the data was assured using several methods. Mass shift was corrected by means of external equilibration with lithium formate clusters at a concentration of 5 mM, dissolved in a solvent of water-isopropanol-formic acid (50:50:0.2 *v*/*v*/*v*), injected with an independent syringe pump before each run, and chromatograms were calibrated in high precision calibration mode with a mass accuracy deviating between 0.01–5 ppm. Furthermore, IS was used to correct the RT of the chromatograms. Blank samples were injected to ensure that carry-over effects did not accrue.

### 4.5. Targeted LC-MS/MS

#### 4.5.1. Materials

The following lignan and flavonoid standards were used for quantitative LC-MS measurements: matairesinol, hydroxymatairesinol, secoisolariciresinol, lariciresinol, isolariciresinol, syringaresinol, medioresinol, pinoresinol, naringenin, formononetin, chrysin, genistein, daidzein, glycitein and prunetin (Plantech, Berksher, UK). The following isotope-labeled and deuterium-labeled internal standards were used: ^13^C_3_-enterolactone and ^13^C_3_-enterodiol (Toronto Research Chemicals, Toronto, ON, Canada) and genistein d4 and daidzein d3 from Cambridge Isotope Laboratories, Inc. (Andover, MA, USA). For the enzymatic hydrolysis, β-glucuronidase type H-1 from *Helix pomatia* was purchased from Sigma (Merck KGaA, Darmstadt, Germany). All solvents used were of HPLC grade. The stock standard solutions were prepared by dilution of purchased lignan standards in acetonitrile and isoflavones in DMSO (Sigma, Merck KGaA, Darmstadt, Germany) and were stored at −80 °C when not in use.

#### 4.5.2. Standard Curve

The standard curve was prepared in 25% acetonitrile in concentrations ranging from 0.0244 to 100 ng/mL, and contained ISs in concentrations of 25 ng/mL for ^13^C_3_-enterolactone, 12.5 ng/mL for ^13^C_3_-enterodiol, 60 ng/mL for daidzein-d3 and 30 ng/mL for genistein-d4. The analyte/internal standard concentration ratio was plotted against the analyte/internal standard peak area ratio as a linear regression curve with 1/x weighting. The quantification of the lignans matairesinol and pinoresinol was performed using ^13^C_3_-enterolactone as an internal standard; and that of hydroxymatairesinol, secoisolariciresinol, lariciresinol, isolariciresinol, syringaresinol and medioresinol using ^13^C_3_-enterodiol as an internal standard. The quantification of isoflavones daidzein and glycitein was performed using daidzein-d3 as an internal standard; and that of genistein, naringenin, formononetin, chrysin and prunitine using genistein-d4 as an internal standard. The lower limit of quantitation (LLOQ) was accepted as the lowest standard on the calibration curve if the analyte response was at least 5 times the response of the blank sample. Data analysis was performed in Analyst software 1.6.2 (AB Sciex, Framingham, MA, USA).

#### 4.5.3. LC-MS/MS and Method Validation

The LC-MS/MS measurements were performed and validated on a microLC 200 series (Eksigent/AB Sciex, Redwood City, CA, USA) and a QTrap 5500 mass spectrometer (AB Sciex, Framingham, MA, USA), according to the method of Nørskov et al. [34,41].

## Figures and Tables

**Figure 1 metabolites-11-00259-f001:**
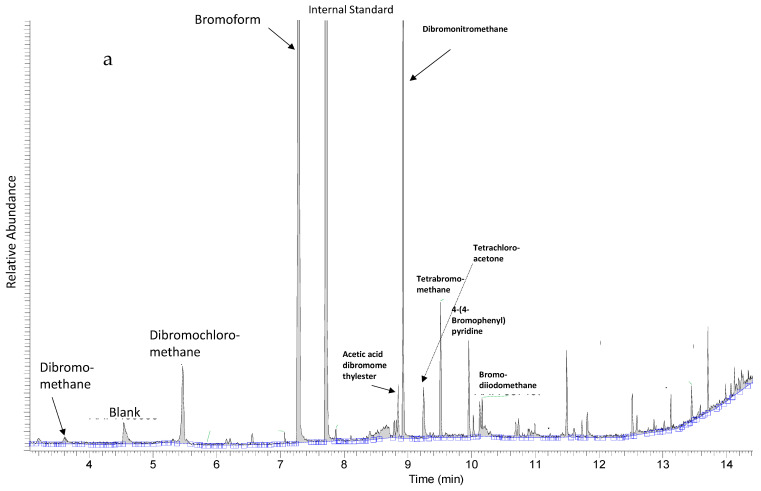
(**a**) Metabolic profile of *Asparagopsis taxiformis* showing halomethanes, bromoform, dibromomethane, dichlorobromomethane and diiodomethane d2, used as an internal standard, and other tentatively identified volatile halocarbons. (**b**) Representative metabolic profile of other seaweeds in this study showing only diiodomethane d2, used as an internal standard.

**Figure 2 metabolites-11-00259-f002:**
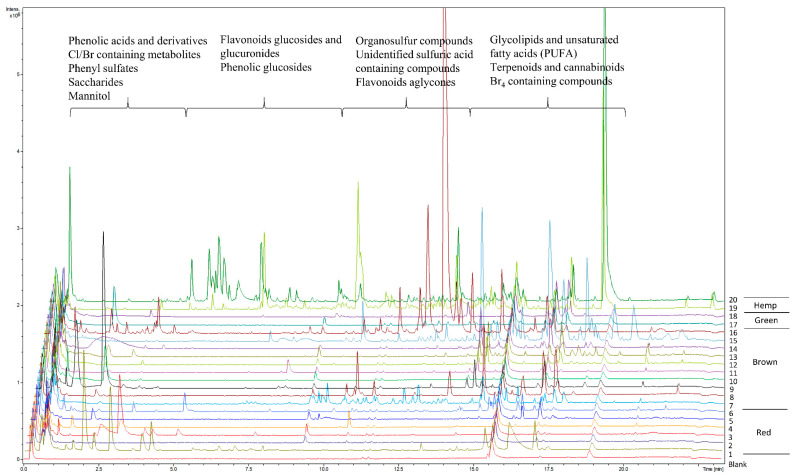
Metabolic profiles of red, brown and green seaweeds and hemp of the *Extract 2* group, measured using untargeted LC-MS. Blank sample represents the extraction solvent used for the extraction of metabolites in samples. Red seaweeds: (1) *Asparagopsis taxiformis*, (2) *Chondrus crispus*, (3) *Furcellaria lumbricalis*, (4) *Polyides rotundus*, (5) *Dumontia contorta*, (6) *Delesseria sanguinea*. Brown seaweeds: (7) *Saccharina latissima*, (8) *Gracilaria vermiculophylla*, (9) *Fucus serratus*, (10) *Fucus spiralis*, (11) *Fucus evanescens*, (12) *Alaria esculenta*, (13) *Laminaria digitata*, (14) *Chorda filum*, (15) *Dictyota dichotoma*, (16) *Sargassum muticum*. Green seaweeds: (17) *Ulva intestinalis*, (18) *Ulva* sp. Hemp: (19) *Cannabis sativa* L. variety Finola, (20) *Cannabis sativa* L. variety Futura 75.

**Figure 3 metabolites-11-00259-f003:**
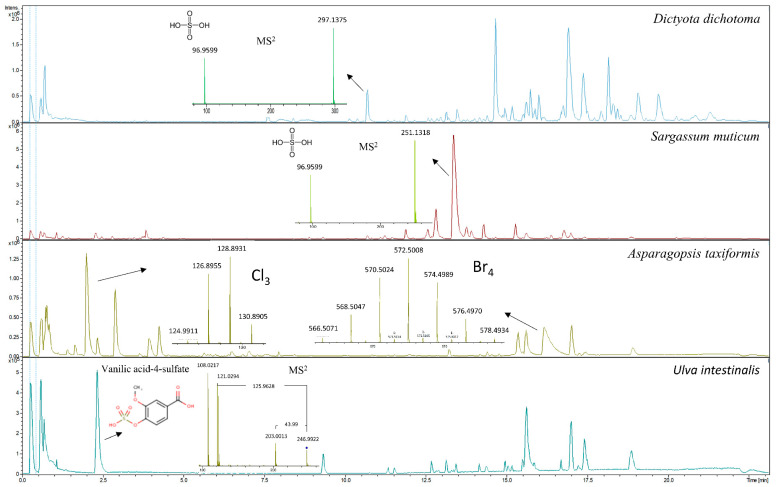
Metabolic profiles of the red seaweed *Asparagopsis taxiformis* showing MS spectra of chlorinated and brominated compounds; metabolic profiles of two brown seaweeds, *Dictyota dichotoma* and *Sargassum muticum*, showing high intensity ions *m*/*z* 297.1375 and *m*/*z* 251.1318, respectively; and the corresponding MS^2^ spectra of an unidentified sulfuric acid containing compounds; and the metabolic profile of the green seaweed *Ulva intestinalis* showing MS^2^ spectra for tentatively identified vanillic acid-4-sulfate.

**Figure 4 metabolites-11-00259-f004:**
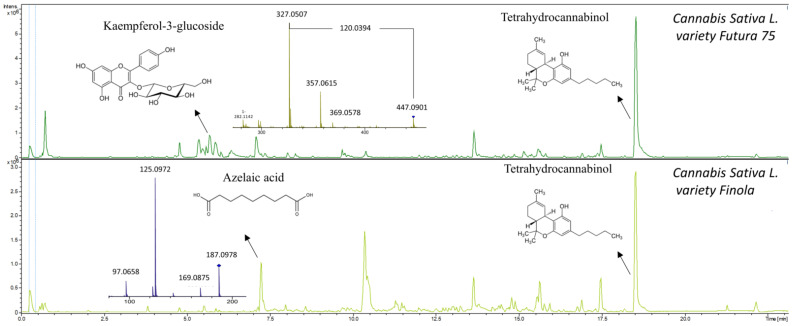
Metabolic profiles of *Cannabis sativa* L. variety Futura 75 and *Cannabis sativa* L. variety Finola showing MS^2^ spectra of kaempferol-3-glucoside and azelaic acid. Tetrahydrocannabinol was also tentatively identified.

**Figure 5 metabolites-11-00259-f005:**
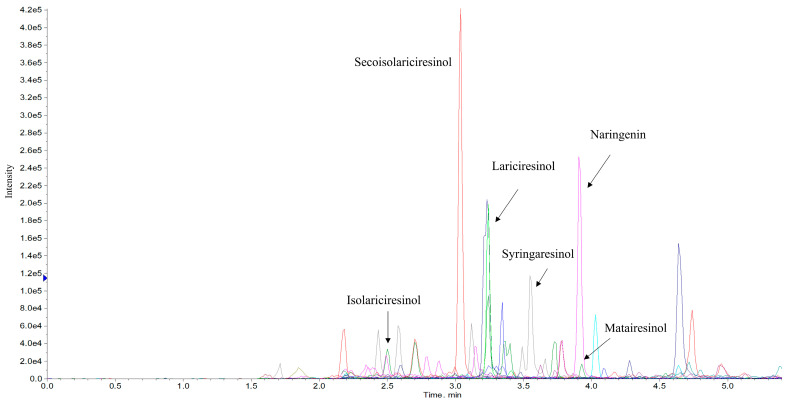
Representative chromatogram of *Cannabis sativa* L. variety Futura 75 Extract 3, showing lignans (lariciresinol, syringaresinol, isolariciresinol, matairesinol and seacoisolariciresinol) and isoflavone (naringenin).

**Figure 6 metabolites-11-00259-f006:**
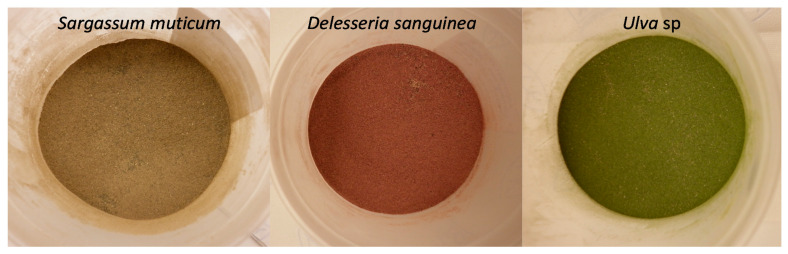
Examples of milled seaweeds. Brown: *Sargassum muticum*, red: *Delesseria sanguinea* and green: *Ulva* sp. seaweeds were milled to pass through a 1-mm screen and homogenized.

**Figure 7 metabolites-11-00259-f007:**
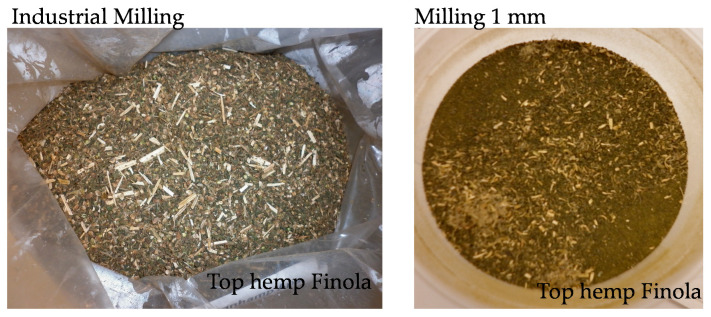
Examples of milled hemp. Top hemp, *Cannabis sativa* L. variety Finola was milled by means of industrial milling and subsequently further milled to pass through a 1-mm screen and homogenized.

**Table 1 metabolites-11-00259-t001:** The concentration of lignans and isoflavones in Extract 2 and Extract 3 per dry weight (DW) of *Cannabis sativa* L. variety Futura 75 and Finola, *n* = 2.

	Futura 75 (µg/100 g)		Finola (µg/100 g)	
	Extract 2	Extract 3	Extract 2	Extract 3
Secoisolariciresinol	4 ± 1.1	235 ± 51	3 ± 0.9	100 ± 12
Lariciresinol	12 ± 1.7	328 ± 20	2 ± 0.3	34 ± 8
Syringaresinol	52 ± 2.8	250 ± 71	27 ± 4.7	81 ± 1.4
Isolariciresinol	3 ± 0.8	39 ± 9	1 ± 0.1	28 ± 1
Matairesinol	0.2 ± 0.01	3 ± 0.2	0.5 ± 0.1	7 ± 4.2
Naringenin	1 ± 0.1	88 ± 31	0.3 ± 0.1	58 ± 15
Glycitein	1 ± 0.3	2 ± 0.4	0.4 ± 0.1	1 ± 0.2

**Table 2 metabolites-11-00259-t002:** Overview of seaweed species used in the analyses. The eighteen species included red, green and brown species harvested in Nordic waters or in Australia from either wild harvesting of natural populations or from kelp species cultivated on longline systems in the Faroe Islands.

Species	Group	Harvest Location	Origin	Harvest Time	Pretreatment
*Asparagopsis taxiformis*	Red	Magnetic Island, AUS	Wild harvest	08-09-2019	Freeze-dried
*Chondrus crispus*	Red	Isefjord, DK	Wild harvest	05-05-2020	Freeze-dried
*Delesseria sanguinea*	Red	Kims top, Kattegat, DK	Wild harvest	08-2020	Freeze-dried
*Dumontia contortia*	Red	Isefjord, DK	Wild harvest	05-05-2020	Freeze-dried
*Furcellaria lumbricalis*	Red	Isefjord, DK	Wild harvest	05-05-2020	Freeze-dried
*Gracilaria vermiculophylla*	Red	Isefjord, DK	Wild harvest	18-02-2020	Freeze-dried
*Polyides rotundus*	Red	Isefjord, DK	Wild harvest	05-05-2020	Freeze-dried
*Ulva intestinalis*	Green	Isefjord, DK	Wild harvest	05-05-2020	Freeze-dried
*Ulva* sp.	Green	Limfjorden, DK	Wild harvest	01-07-2020	Freeze-dried
*Alaria esculenta*	Brown	Funningsførdur, Faroe Isl.	Cultivation	Spring 2020	Oven-dried 40 °C
*Chorda filum*	Brown	Isefjord, DK	Wild harvest	15-09-2020	Freeze-dried
*Dictyota dichotoma*	Brown	Limfjorden, DK	Wild harvest	01-09-2020	Freeze-dried
*Fucus evanescens*	Brown	Isefjord, DK	Wild harvest	05-05-2020	Freeze-dried
*Fucus serratus*	Brown	Isefjord, DK	Wild harvest	05-05-2020	Freeze-dried
*Fucus spiralis*	Brown	Isefjord, DK	Wild harvest	05-05-2020	Freeze-dried
*Laminaria digitata*	Brown	Funningsførdur, Faroe Isl.	Cultivation	Spring 2020	Oven-dried 40 °C
*Saccharina latissima*	Brown	Funningsførdur, Faroe Isl.	Cultivation	Spring 2020	Oven-dried 40 °C
*Sargassum muticum*	Brown	Limfjorden, DK	Wild harvest	01-09-2020	Freeze-dried

DK: Denmark. AUS: Australia. Faroe Isl.: Faroe Islands.

**Table 3 metabolites-11-00259-t003:** Quantitative and qualifying ions, retention time (RT) and molecular weight (MW) of halomethanes.

Compounds	MW (g/mol)	RT (min)	Selected Ion (*m*/*z*)			
			Quantitative ion	Ion clusters	Qualifying ion	Ion clusters
Dibromomethane	173.83	3.6	174	172, 174, 176	93	91, 93, 95
Dibromochloromethane	208.28	5.5	129	127, 129, 131	-	-
Bromoform	252.73	7.3	173	171, 173, 175	252	250, 252, 254, 256
Diiodomethane-d2	269.84	7.7	270	-	143	-

**Table 4 metabolites-11-00259-t004:** Recovery % and precision (RSD) of the method.

Compounds	Spiked 1 µg/mL (LLOQ)		Spiked 5 µg/mL		Spiked 50 µg/mL	
	Recovery %	Precision RSD %	Recovery %	Precision RSD %	Recovery %	Precision RSD %
Dibromomethane	120	4.0	120	5.1	118	2.0
Dibromochloromethane	91	5.2	99	7.5	103	2.3
Bromoform	112	3.9	101	6.3	88	2.2

## Data Availability

The data presented in this study are available on request from the corresponding author.

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
