# Peer review of "Targeted and Untargeted Metabolic Profiling to Discover Bioactive Compounds in Seaweeds and Hemp Using Gas and Liquid Chromatography-Mass Spectrometry"

_metabolites, 2021, doi:10.3390/metabo11050259_

Round 1
Reviewer 1 Report
The article is interesting and deals with a very current topic.
- In the "Introduction" page 1,lines 34-39: I suggest to indicate some references to support the sentences.
- Line 183, table 1: results are reported as average values of two replicates. I suggest to specify (in par. 4) whether the replicates are from two independent extractions orare two injections of the same extract.
- In the "Materials and methods" section 4.2, the authors reported they use the enzyme beta glucuronidase in acetate buffer 2mg / mL; and in section 4.4, they say the enzyme is purchased from Sigma: I suggest to indicate also the enzymatic activity in terms of Units / mL or Units / mg.
- There are few typos/spelling errors:
- Line 40: please change "experiment have shown" to "experiments have shown" or "experiment has shown".
- Line 52: please check "Dictyota bartayresii was showen to inhibite methane by over 92%.".
- Line 136: it should be "p-hydroxybenzaldehyde" instead of "p-hydroxybenaldehyde".
- Line 156: it should be "gallocatechin" instead of "gallacatechin".
- Line 250: it should be "therefore" instead of "therefor".
- Line 328-329: please check the sentence "Furthermore, metabolic profiling revealed unique chemistry of Asparagopsis taxiformis containing volatile halocarbons, which has is known to have high methane reductive potential".
Author Response
- The references has been added
- The suggested specification was added in par. 4.2
- The suggested specification was added in par. 4.2
- The typing/spelling errors were corrected
Reviewer 2 Report
- Abstract can be more concise.
- Halomethans OR Halomethanes in abstract; gasses in keywords & introduction requires attention, in case, these are typos.
- m/z, p should be in italics
- Conclusion can be separated from discussion.
- h or hr can be used for hours
- ref1, 4-7, 13, 27-30, 37 has no page numbers
- Also some references are missing issue numbers e.g. ref13, 27
- Ref16 requires correction in article number, journal name and authors (I.A.)
- Doi numbers need to mentioned for all references.
Author Response
- As it is suggested by the journal the abstract contains: (1) Background: Place the question addressed in a broad context and highlight the purpose of the study; Greenhouse gas emissions are a global problem facing the dairy/beef industry. Novel feeds additives, seaweeds and hemp containing bioactive compounds are theorized to reduce enteric methane emission. This study aimed to investigate the metabolic profiles of brown, red and green seaweeds and hemp plant using Gas Chromatography- and Liquid Chromatography-Mass Spectrometry. (2) Methods: briefly describe the main methods or treatments applied; We used targeted and untargeted approaches, quantifying known halomethanes and phenolics as well as identifying potentially novel bioactive compounds with anti-methanogenic properties. (3) Results: summarize the article's main findings; The main findings were: a) Asparagopsis taxiformis contained halomethanes with high concentration of bromoform (4200 µg/g DW) and six volatile halocarbons were tentatively identified, b) no halomethanes were detected in other studied seaweeds nor hemp, c) high concentrations of lignans were measured in hemp, d) high numbers of sulfated phenolic acids and unidentified sulfuric acid containing compounds were detected in seaweeds, e) flavonoid glucosides and glucuronides were mainly identified in hemp, f) condensed tannin, gallocatechin was tentatively identified in Fucus sp. (4) Conclusions: indicate the main conclusions or interpretations; Using combined metabolomics approach, overview and in depth information on secondary metabolites were achieved. Halomethans of Asparagopsis sp. have already been shown to be anti-methanogenic, however, metabolic profiles of seaweeds such as Dictyota and Sargassum have also shown to contain compounds that might have anti-methanogenic potential.
Due to particular requirements by the journal, we think that we have to provide the necessary level of details. The abstract is within 200 words.
- The spelling of methane and halomethanes has been checked
- revised as requested
- Separate conclusion is not required by the journal. Discussion section is already quite long since many different results have to be discussed and it will be challenging to write concise conclusion without to much repetition.
- revised as requested
- not all the references contain information on page numbering
- -9. revised as requested
Reviewer 3 Report
In general the manuscript is well written, the methodology well described, etc.
However, I don't quite understand that algae and hemp are used in the same work. It is logical that no lignans are found in algae and no halomethanes are found in hemp. The fact that both are used as antimethanogenic additives does not seem to me to be sufficient motivation.
I propose to the authors to divide the work according to the species analyzed.
Author Response
We understand the concern of the reviewer, however in the study of Zhong et al several lignans were identified, deoxyschisandrin, dimethylmatairesinol, arctigenin and 2-hydroxyenterolactone. These lignans were identified in Glateloupia sp., Centroceras sp., Sargassum sp., Caulerpa sp., Dasya sp., Ecklonia sp., and Codium sp. That indicates that seaweeds can contain lignans, though they are probably not the most abundant compounds in seaweeds. We have performed targeted analyses, though different lignans then in the study of Zhong et al, and did not detected. However, it is important to measure before one can say whether these compounds present there or not.
This study investigates bioactive compounds for particular purpose, anti-methanogenic properties. Bioactive compounds with anti-methanogenic properties can come from variety of sources. There are studies that investigate both different grasses and trees, because they are good sources of bioactive compounds. We have added a section in the introduction, which we hope will improve the understanding why the seaweeds and hemp are used in the same work.
Round 2
Reviewer 3 Report
I think that with the modifications made in the introduction the purpose of this work is clear. I think it is suitable for publication.